# EsReflux Protocol: Epidemiological Study of Heartburn and Reflux-like Symptoms in Spanish Community Pharmacies

**DOI:** 10.3390/ijerph19169807

**Published:** 2022-08-09

**Authors:** Elsa López-Pintor, María Puig-Moltó, Blanca Lumbreras

**Affiliations:** 1Department of Engineering, Area of Pharmacy and Pharmaceutical Technology, Miguel Hernandez University, 03550 San Juan de Alicante, Spain; 2CIBER of Epidemiology and Public Health, CIBERESP, 28029 Madrid, Spain; 3Department of Public Health, History of Science and Gynecology, Miguel Hernandez University, 03550 San Juan de Alicante, Spain

**Keywords:** heartburn, reflux, dyspepsia, community pharmacy, intervention, reflux-like symptoms

## Abstract

(1) Background: Heartburn and reflux discomfort are frequent reasons for consultation at the community pharmacy. To facilitate the assistance work of the community pharmacist and its coordination between different levels of care, a group of experts in Community Pharmacy, Primary Care, and Gastroenterology have recently worked on an algorithm to manage these symptoms in the community pharmacy (Professional Pharmaceutical Service). The objective of this study is to analyze the clinical and sociodemographic characteristics of patients with heartburn and/or reflux-like symptoms who go to a community pharmacy, and to evaluate the clinical and humanistic results after the implementation of a Professional Pharmaceutical Service. (2) Methods: A pre-post study will be carried out to evaluate clinical and humanistic results after the implementation of a Professional Pharmaceutical Service. We will include 1200 patients who ask for advice or get a non-prescription medication due to acid and/or reflux symptoms in 240 Spanish pharmacies. Clinical data will be collected at baseline and 15 days after the pharmaceutical intervention. The GERD Impact Scale (GIS) questionnaire will be applied to assess changes in heartburn/reflux-like symptoms and quality of life after the intervention.

## 1. Introduction

Gastroesophageal reflux, the passage of gastric content into the esophagus, presents as the main symptoms of heartburn and/or reflux. Heartburn associated with reflux is suffered postprandially in many patients as a painful retrosternal burning sensation [1,2]. Treatment with an alginate-antiacid combination has shown effectiveness with the displacement neutralization of the acid pocket [3,4]. However, heartburn symptoms can be also diagnosed in other affections, such as dyspepsia, defined as pain or discomfort at the epigastric region (epigastric pain syndrome) and/or postprandial fullness, early satiety (postprandial distress syndrome) related to the digestive area. Antiacid therapy has shown effectiveness when the patient has sporadic symptoms [5,6,7]. The recognition of symptoms and the location (epigastric and/or retrosternal) of the different gastrointestinal affections is crucial for the early diagnosis of diseases that could develop more serious complications, such as gastro-oesophageal reflux disease (GERD), Barrett’s oesophagus, etc., and the specific management of the symptoms.

The prevalence of these symptoms varies according to epidemiology and sociodemographic characteristics such as geographic location, age, sex, body mass index (BMI), nutrition habits, etc., [8,9,10,11]. Heartburn is reported by 20−40% of subjects in the Western population [9] and according to a meta-analysis, the prevalence of dyspepsia is 21% [10]. A recent meta-analysis has estimated that the worldwide prevalence of one of the main complications, GERD, is 13.3% [12], and in Spain, the prevalence is 9.8% but with variations between different geographic areas [8]. Although most subjects report that the duration of these symptoms are limited, approximately 70% of them report symptoms during two or more days per week [11]. The impact of these symptoms on work activity is also relevant, with a high frequency of absenteeism and reduction in the number of productive hours, which implies significant economic losses, both due to direct costs and those related to medical visits and treatments [13].

Heartburn and reflux-like symptoms are frequent reasons for consultation at the community pharmacy and the characterization of them is often complicated, and frequently, the community pharmacy will be the only point of consultation for patients. Therefore, it is necessary to know the clinical and sociodemographic characteristics of the patients who go to the community pharmacy with heartburn and/or reflux, as support to characterize the problem and to develop preventive and management strategies, including referral to Primary Care when it was needed.

To improve the coordination between the community pharmacist and the different levels of health care, a group of experts in Community Pharmacy, Primary Care, and Gastroenterology has recently worked on an algorithm to manage these symptoms, with the support of the Spanish Society Community Pharmacy (SEFAC) and the Spanish Society of Primary Care Physicians (SEMERGEN) [14]. This algorithm establishes the management strategies for a pharmacist (Professional Pharmaceutical Service) when a patient with heartburn and/or reflux-like symptoms goes to a community pharmacy, after the classification of the patient according to the location of symptoms (epigastric and/or retrosternal) and the degree of severity of symptoms. Previous algorithms have been developed in the diagnosis and management of patients with heartburn and/or reflux-like symptoms [15]. In 2002, Gstaad Treatment Guidelines, an algorithm for the management of GERD was developed [16]. In 2008, this algorithm was updated to provide clear guidance for all healthcare professionals in order to better manage this pathology, enabling pharmacists to use it as well as doctors [17]. This algorithm focused on the treatment of GERD with proton pump inhibitor (PPi) and it did not take into account minor symptoms. Although a group of gastroenterological and pharmaceutical experts participated in this update, the algorithm did not reflect daily practice in a community pharmacy in which both patients with minor and serious symptoms seek advice. Thus, the algorithm validated in our study aims to include the characteristics of patients with serious and minor symptoms in a community pharmacy setting.

This study aims to analyse the epidemiological characteristics of patients with heartburn and/or reflux-like symptoms who attend Spanish community pharmacies and evaluate the impact of a Professional Pharmaceutical Service in both patient’s symptoms and quality of life.

## 2. Materials and Methods

### 2.1. Study Design

A pre-post interventional study will be carried out to evaluate the impact of a Professional Pharmaceutical Service in both patient’s symptoms and quality of life.

### 2.2. Setting

The study will be carried out throughout the national territory, with an estimated participation of 240 community pharmacies. The inclusion of participants and data collection will take place between January and June of the year 2023. There are approximately 22,000 community pharmacies in Spain with an average of 2117 people per pharmacy, the lowest ratio in the European Union after Greece 15 (the minimum population allowed for opening a new pharmacy range from 700 to 2500 in states). In Spain, Law 16/1997, of 25 April 1997, regulating the services provided by pharmacies, establishes that pharmacies are private health establishments of public interest, subject to the health planning established by the Autonomous Communities. This Law establishes that in Spain, the number of inhabitants per pharmacy is around 2200 [18]. This means that on average, one pharmacy has to be supplied for every 2117 patients. Spain is one of the European countries with the highest number of pharmacies per capita. All community pharmacies are privately owned with only pharmacists owning a single community pharmacy, although more than one pharmacist may jointly own a pharmacy.

### 2.3. Study Population

We will include patients ≥ 18 years who attend a community pharmacy for symptoms related to heartburn and/or reflux, or ask for treatment to improve them. We will exclude subjects who ask for treatment for another person or women with a risky pregnancy.

The study participants will be considered included after signing the corresponding informed consent.

### 2.4. Sample’s Size Calculation

The sample size determination is based on the second of the objectives in which we evaluate the impact of a Professional Pharmaceutical Service in both patient’s symptoms and quality of life. There is no previous data on the management of patients with symptoms of heartburn and/or reflux at the community pharmacy. However, previous studies have been carried out in other clinical settings in patients with reflux symptoms evaluated through the Gastroesophageal Reflux Disease Impact Scale (GIS).

We estimated the number of people to assess 0.1 units of change over a 4-point Likert scale between 2 visits (from 2.2 to 1.1) with a SD of 0.6 through the estimation of the mean in repeated measures. We considered an alpha risk of 0.05, two-sided test and a beta risk of 0.20 with a dropout of 20% [19]. With these calculations, we estimated to be needed to include 707 patients. However, given that we want to analyse different factors associated to this change (age, sex, race, educational level; b) clinical data: obesity, food and nutrition habits (coffee, chocolate, tea, tomato, spicy food, citrus, and carbonated drinks), physical exercise, smoking and alcohol habits, family history, and other gastrointestinal problems, previous and current medication we will increase the sample size until 1200 patients (15 patients per variable).

### 2.5. Recruitment Procedure

With the aim of obtaining a greater possible representation of the Spanish territory, and thus, achieve a more accurate description of the clinical and demographic symptoms of patients who attend a community pharmacy, we estimate that each community pharmacy will include a minimum of 5 patients until the desired sample size is reached. Therefore, and considering the necessary sample size (around 1200 people), we would need to include 240 community pharmacies. We had a list of consecutive pharmacies which were able to participate in the study. Therefore, if a pharmacy declined to participate, we contacted another pharmacy.

The Spanish Society of Community Pharmacy (SEFAC) will offer the participation to the nearly 6000 community pharmacists’ members to the society. The participation of each community pharmacy will be randomly selected, considering the population of each of the autonomous communities, until the required sample size is reached (240 community pharmacies).

All registered pharmacists will receive prior training by the research team on the objective, methodology and procedures of the study. This training will be accredited by the Miguel Hernández University of Elche, where the research team is located. The content of the training will be available to pharmacists throughout the duration of the study on a specific web platform of the EsReflux project. This training will consist of three training videos with a duration of approximately 2 h, explaining the physiology and descriptions of symptoms and how to use the algorithm and the web platform. In addition, the external monitor will visit each pharmacist to test that the pharmacist has understood the procedure of the study.

### 2.6. Data Collection Procedure

The Esreflux project will include two visits, at baseline and 14 days after the intervention (Figure 1).

### 2.7. Baseline Visit

Phase 1: Those patients who fulfil the selection criteria will be invited to participate in the Esreflux project. The community pharmacist will explain the study and if patient agrees, after signing the informed consent will be included in the study. Each participant will be consecutively anonymised with an identification code as follows: CA-III-PN (CA: autonomy code; III: initials of the researcher; PN, and N the number of participant).

The community pharmacist will collect the following data through a Data Collection Notebook (CDR): (a) sociodemographic data: age, sex, race, educational level; (b) clinical data: obesity, food and nutrition habits (coffee, chocolate, tea, tomato, spicy food, citrus, and carbonated drinks), physical exercise, smoking and alcohol habits, family history, and other gastrointestinal problems, previous and current medication. In addition, the pharmacist will advise the patient in case of concerning symptoms.

Previously to the application of the intervention, the community pharmacist will evaluate the heartburn/reflux-like symptoms (duration/intensity) through the application of the GERD Impact Scale questionnaire (GIS questionnaire). This is a questionnaire validated in Spain [20] for the healthcare professional to manage the patient with reflux. The GIS is composed of nine questions and uses a four-graded Likert scale for answers (i.e., daily, often, sometimes, and never) (Table 1). The recall period for the questions was seven days after the study visits. The nine questions cover three dimensions: upper GI symptoms (questions 1a, 1b and 1d), other acid-related GI symptoms (questions 1c and 1e) and the impact of the symptoms on the patient’s daily lives (questions 2, 3, 4 and 5). A mean score will be calculated for each dimension, generating a number between 1 and 4. In addition, the pre–post changes from Visit 1 to Visit 2 will be calculated within each severity level.

In addition, its usefulness has been seen in the follow-up of the patient [16] since it includes aspects of reflux symptoms and quality of life, so it can be applied to the evaluation of changes in patient management.

An independent external monitor with clinical trial experience will regularly monitor compliance with the study protocol in each community pharmacy and validate each patient’s questionnaire prior to enrolment in the study.

Phase 2: Once the data have been collected, the community pharmacist will apply the algorithm [14] for patients with reflux-like symptoms developed by the different scientific societies. This algorithm establishes the different steps to be followed by the community pharmacist in communication with the primary care physician (Figure 2).

Phase 3. Follow-up visit: Evaluation of results. Fourteen days after the pharmaceutical intervention is carried out, the independent external monitor will phone the patient to collect data related with the efficacy of the intervention and the patient’s satisfaction with the pharmaceutical care received.

### 2.8. Outcome Measures

In the follow-up visit, pharmacists will assess the evaluation of the patient’s heartburn/reflux-like symptoms (duration/intensity) and their quality of life through the GERD Impact Scale Questionnaire (GIS) [19,20]. We will consider that the Professional Pharmaceutical Service has an impact if there is a significant difference in the GIS Questionnaire score for one of the three dimensions (upper GI symptoms, other acid-related GI symptoms and the impact of the symptoms on the patient’s daily lives) before and after the intervention. We will also assess the change in the overall score (change of 0.1 compared with the previous score).

We carried out a systematic review of those reviews which evaluated the available instruments for assessing the patient-reported outcomes for GERD symptoms. Based on the articles retrieved [21,22,23,24,25] we decided to include the GERD Impact Scale Questionnaire (GIS) for the evaluation of patient’s heartburn/reflux-like symptoms.

The monitor will also evaluate patients’ satisfation with pharmacist intervention through the Armando Patient Satisfaction Questionnaire [26] and with the medication (for those patients who received a pharmacological treatment) through the Treatment Satisfaction Questionnaire for Medication [27].

### 2.9. Data Analysis Plan

The data analysis will be carried out by a Data Monitoring Committee (DMC) formed by the research team (principal investigator, collaborating researcher and external monitor) of the Miguel Hernández University of Elche, independent of the sponsor and competing interests.

All data will be collected in the CDR and only the principal investigator and the external monitor will have access. The data collected will be included in a structured database according to the variables selected for the study. The data will be checked for correct entry by the monitor. The analysis will be carried out with IBM SPSS Statistics for Windows, Version 27.0. Armonk, NY, USA: IBM Corp.

Data will be presented in descriptive form with specification of absolute and relative frequencies. We will calculate the precision of each measure by determining the 95% confidence interval.

To assess the impact of the intervention, and given that the questionnaire that is applied has 9 items (5 of symptoms and 4 of quality of life) with 4 scales in each (from daily to never), we will use the 4-point Likert scale, and a *p*-value less than 0.05 will be considered significant.

### 2.10. Ethics

This study has been approved by the Ethical Committee of the Sant Joan D’Alacant Hospital and by the Spanish Medicine Agency (AEMPS).

Participation in the study by research pharmacists will be voluntary, and independent. Research pharmacists will receive compensation proportional to the time and additional responsibilities dedicated to the study. This financial compensation will be explicit and transparent and will be made known to the evaluating Institutional Review Board (IRB)/Independent Ethics Committee (IEC).

The research pharmacist will sign a researcher commitment by means of which he/she to correctly collect validated real data, which may be useful and valuable in the research. The research pharmacist will inform the patient about his/her inclusion in the study, both orally and in writing, through the Participant Information Sheet, ensuring that he/she understands the information provided before giving their consent and that he/she can leave the research study at any time.

The data provided by the patient will be treated confidentially, according to the General Data Protection Regulation (Regulation 2016/679 of 27 April). Only the external monitor and the community pharmacist will have access to the personal data such as telephone number (for the follow-up visit) and the main symptoms and treatments.

In the database, patients will be individualized by means of dissociated, non-identifiable, nonsense codes for any other information system and that will not allow the identification of individual patients or their crossing with other databases (each participant will be consecutively anonymized with an identification code as follows: CA-III-PN (CA: autonomy code; III: initials of the researcher; PN, and N the number of participant)). The project database will not contain any data that allows the identification of patients, and the research team will not have-neither from these databases nor from other sources-patient identification information.

### 2.11. Dissemination

Results will be published in scientific journals and presented at congresses of societies such as the Spanish Society Community Pharmacy (SEFAC) and reports will be shared on pharmacist platforms. Authorship eligibility will be made following the International Committee of Medical Journal Editors Recommendations for the Conduct, Reporting, Editing, and Publication of Scholarly Work in Medical Journals (ICMJE Recommendations 2018). The use of professional writers is not foreseen.

## 3. Discussion

Knowing patients’ clinical and sociodemographic variables associated with heartburn and reflux-like symptoms will allow community pharmacists to earlier detect those patients with a higher risk of complications, and therefore, which patients need to be referred to the primary care. In addition, the evaluation of the impact of the Pharmaceutical Professional Service on in both patient’s symptoms and quality of life will show the effectivity of including such service in the community pharmacies to improve the management of affected patients.

Previous studies carried out in community pharmacies showed positive results. Previous research [28] showed how pharmacist-provided medication therapy management services effectively reduced costs associated with patient medication use in patients with GERD. Another study carried out in Swedish pharmacies [29] demonstrated a counseling model designed to discover and resolve problems related to symptoms, and drug use appeared to have a favorable impact on outcomes in customers with dyspepsia seeking nonprescription drug treatment in Swedish pharmacies. Nevertheless, this is, to our knowledge, the first study that assess the characteristics of patients attending to community pharmacies for heartburn/reflux-like symptoms, and the impact of a Pharmaceutical Professional Service in collaboration with Primary Care to manage these patients.

This study is not without limitations. Firstly, the Pharmaceutical Professional Service is based on patients’ self-reported symptoms, and given the heterogeneity of symptoms in these pathologies, it may be difficult to differentiate. However, the inclusion of different questions about their symptomatology and the validated Questionnaire GIS will help the pharmacist to improve the detection of the symptoms. Secondly, some patients may be lost to follow-up; however, given the short period of time between the two visits, we expect these losses to be minimal. In any case, it will provide us with information about the epidemiological characteristics of these patients. Lastly, we will only assess the adherence to treatment in those patients who are given a new medication to deal with their symptoms. However, according to the algorithm, not all the patients will be given treatment. We will assess the patients’ satisfaction with the intervention and the difficulties to follow the recommendations, so, we will be able to indirectly assess if they follow them. We have included this explanation in the limitations section.

## 4. Conclusions

This study will allow us to analyse the epidemiological characteristics of patients with heartburn and/or reflux-like symptoms who attend Spanish community pharmacies. This will allow community pharmacists to detect early those patients with a higher risk of complications and, therefore, which patients should be referred to primary care. In addition, evaluation of the impact of the Professional Pharmacist Service on both symptoms and quality of life of patients will show the efficacy of including such a service in community pharmacies to improve the management of affected patients. 

## Figures and Tables

**Figure 1 ijerph-19-09807-f001:**
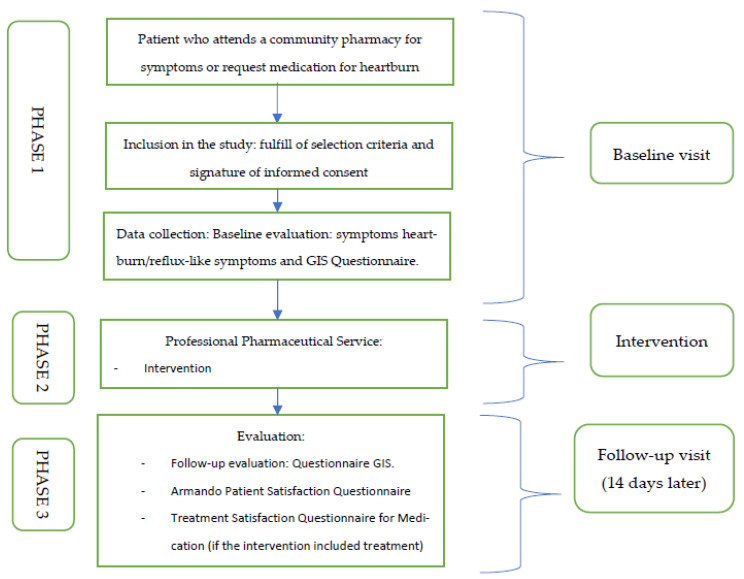
Description of the study procedure.

**Figure 2 ijerph-19-09807-f002:**
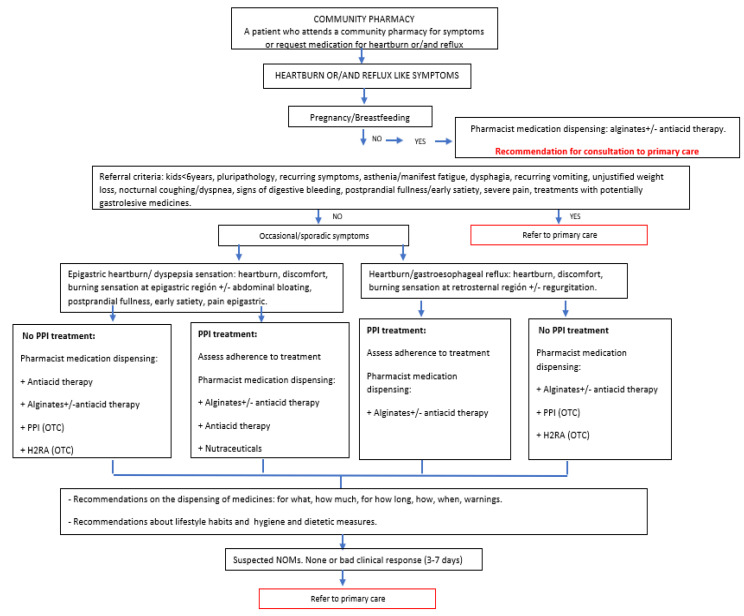
Heartburn and/or reflux-like symptom management algorithm in community pharmacy.

**Table 1 ijerph-19-09807-t001:** Questions of the GERD Impact Scale (GIS).

1. How often have you had the following symptoms: a. Pain in your chest or behind the breastbone? b. Burning sensation in your chest or behind the breastbone? c. Regurgitation or acid taste in your mouth? d. Pain or burning in your upper stomach? e. Sore throat or hoarseness that is related to your heartburn or acid reflux? 2. How often have you had difficulty getting a good night’s sleep because of your symptoms? 3. How often have your symptoms prevented you from eating or drinking any of the foods you like? 4. How frequently have your symptoms kept you from being fully productive in your job or daily activities? 5. How often do you take additional medication other than what the clinician told you to take?

## Data Availability

Data sharing is not applicable to this protocol as no new data were created or analyzed in this study.

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
