# Peer review of "EsReflux Protocol: Epidemiological Study of Heartburn and Reflux-like Symptoms in Spanish Community Pharmacies"

_ijerph, 2022, doi:10.3390/ijerph19169807_

Round 1
Reviewer 1 Report
This manuscript is a protocol for a pre-post intervention study. I don't think this type of manuscript if of interested to the readers of IJERPH.
Author Response
Dear Reviewer,
Many thanks for your time. The specific response to your comments is highlighted as follows.
This manuscript is a protocol for a pre-post intervention study. I don't think this type of manuscript if of interested to the readers of IJERPH.
We respect and sincerely appreciate your opinion. However, as we made clear in our first response, protocols are among the types of articles that IJERPH accepts, and furthermore, in our opinion, the publication of research protocols such as the one presented here, under real conditions, is essential to promote transparency and quality research in routine clinical practice. At the same time, the scientific literature does not include many protocols of work carried out in the field of community pharmacy, in which the Scientific Societies and the industry also participate, as well as a multidisciplinary team, a very important synergy for the generation of evidence in the real world. Therefore, we sincerely believe that our manuscript will be of interest to IJERPH readers.
Reviewer 2 Report
There might be some style and/or grammatical changes needed (eg Line 220 and 'receding'), but those are easily fixed during the editorial process. For me, they addressed what I had issue with.
Author Response
Dear Reviewer,
We sincerely appreciate your careful reading of the manuscript and your constructive comments and observations, which helped us to improve the quality of the article.
The specific responses to your comment is highlighted as follows. Many thanks for your time!
*****************************************************************************
- There might be some style and/or grammatical changes needed (eg Line 220 and 'receding'), but those are easily fixed during the editorial process.
Done, thank you very much. We have made an extensive revision of the manuscript and have corrected stylistic and grammatical errors, such as the one you have pointed out (page 5, lines 201-202).
This manuscript is a resubmission of an earlier submission. The following is a list of the peer review reports and author responses from that submission.
Round 1
Reviewer 1 Report
I will attach my thoughts on this.
Thanks for setting up this project to help understand heartburn in community pharmacies, where I work and see a lot of such patients.

Reviewer 2 Report
Dear Authors,
The project of presented study seems to be interesting, and I hope to see the results after the study will have been conducted. However, the manuscript needs some improvements.
I would suggest rethinking the structure of the article. I do not fully understand the idea of presenting future project of study as a research article. Some sections should be removed from the manuscript because they are not logical. For instance, there is no need to mention the ‘Discussion’ in the abstract, especially if there are no results to be discussed.
The same incompatibility occurs in the body of manuscript. The main goal of the ‘Discussion part’ is to compare the results to other studies. In case there are no results yet – what can it be compared to and how discussed?
When reading ‘Sample size calculation’ and description of ‘Baseline visit’ some numbers can be found in lines: 110, 153, 154. What do mentioned numbers mean?
Figure 2 is impossible to be read, please improve it.
It is not reasonable to include ‘Conclusions’ part if there are no results. What do authors conclude? (the same as with ‘Discussion’)
Regards,
Reviewer 3 Report
This manuscript describes a protocol for conducting an interventional study to assess its impact.
IJERPH Instruction for authors publication types does not include protocols as studies accepted for publication. Original research manuscripts have to provide data on analyzed data and provide a substantial amount of new information, which is not the case here.